# The Effect of Interface Diffusion on Raman Spectra of Wurtzite Short-Period GaN/AlN Superlattices

**DOI:** 10.3390/nano11092396

**Published:** 2021-09-14

**Authors:** Valery Davydov, Evgenii M. Roginskii, Yuri Kitaev, Alexander Smirnov, Ilya Eliseyev, Eugene Zavarin, Wsevolod Lundin, Dmitrii Nechaev, Valentin Jmerik, Mikhail Smirnov, Markus Pristovsek, Tatiana Shubina

**Affiliations:** 1Ioffe Institute, St. Petersburg 194021, Russia; e.roginskii@mail.ioffe.ru (E.M.R.); yu.kitaev@mail.ioffe.ru (Y.K.); alex.smirnov@mail.ioffe.ru (A.S.); ilya.eliseyev@mail.ioffe.ru (I.E.); ezavarin@mail.ioffe.ru (E.Z.); lundin.vpegroup@mail.ioffe.ru (W.L.); nechayev@mail.ioffe.ru (D.N.); jmerik.pls@mail.ioffe.ru (V.J.); shubina@beam.ioffe.ru (T.S.); 2Faculty of Physics, Saint-Petersburg State University, St. Petersburg 199034, Russia; m.smirnov@spbu.ru; 3Institute for Materials and Systems for Sustainability, Nagoya University, Nagoya 464-8601, Japan; pristovsek@imass.nagoya-u.ac.jp

**Keywords:** GaN/AlN superlattices, molecular beam epitaxy, metal-organic vapor phase epitaxy, random-element isodisplacement model, density functional theory, lattice dynamics, group theory analysis, Raman spectroscopy, interface diffusion

## Abstract

We present an extensive theoretical and experimental study to identify the effect on the Raman spectrum due to interface interdiffusion between GaN and AlN layers in short-period GaN/AlN superlattices (SLs). The Raman spectra for SLs with sharp interfaces and with different degree of interface diffusion are simulated by *ab initio* calculations and within the framework of the random-element isodisplacement model. The comparison of the results of theoretical calculations and experimental data obtained on PA MBE and MOVPE grown SLs, showed that the bands related to *A*_1_(LO) confined phonons are very sensitive to the degree of interface diffusion. As a result, a correlation between the Raman spectra in the range of *A*_1_(LO) confined phonons and the interface quality in SLs is obtained. This opens up new possibilities for the analysis of the structural characteristics of short-period GaN/AlN SLs using Raman spectroscopy.

## 1. Introduction

The functioning of modern optoelectronic and electronic devices based on quantum-sized heterostructures in a system of (Al, Ga)N materials critically depends on accurate control of the composition of the layers and the quality of interfaces between them. Control can be improved by replacing ternary compound layers with extended short-period superlattices (SLs) of binary GaN/AlN compounds with layer thicknesses varied with submonolayer accuracy (1 monolayer (ML) = 0.25 nm), both in active and emitter regions (with *p*- and *n*-type conductivity) of device structures. This approach automatically ensures the spatial homogeneity of the composition, as well as atomically smooth morphology. Its application can increase the quantum efficiency of light-emitting structures, or achieve required doping level, etc. Therefore, extended GaN/AlN SLs have been included in various photodetecting and light-emitting infrared devices based on intersubband transitions [1]. Quantum wells (QWs) based on layers of GaN compounds in a wider gap AlN matrix and GaN/AlN SLs are considered for use in the architectures of many optoelectronic devices in the ultraviolet (UV) range as active regions [2] or reflective Bragg mirrors [3]. Many problems in the production of high-performance UV lasers and light-emitting diodes can be solved, *inter alia*, through the use of extended GaN/AlN SLs (also called digital solid solutions, or digital alloys), which provide doping of the emitter layers with a high average Al content (>40 mol%) [2,4]. A significant contribution to the development of nitride microwave electronics can be made by resonant microwave diodes based on GaN/AlN QWs [5], and microwave transistors based on GaN/AlN SLs [6].

All of the above applications require a high quality of the interfaces between GaN and AlN, i.e., sharp upper and lower heterointerfaces, as well as atomic smoothness with a standard deviation of the surface roughness at the level of 1–2 MLs over a few square microns.

The quality of interfaces in GaN/AlN heterostructures obtained using various technologies is determined by many factors, including, first of all, the surface mobility of precursors (adatoms) and the rate of altering incorporation. This does not only depend on supply, but also involves surface processes, like metal overlayer [7,8,9]. Another important factor for the sharpness of heterointerfaces is the atomic segregation either to the surface of structures or to internal defects (threading dislocations). In addition, the smoothness of surfaces in heterostructures can significantly depend on the processes of generation and relaxation of elastic stresses arising both during the initial stages of growth of buffer layers on the substrates with different crystallographic mismatches and the sequential growth of layers.

New non-destructive methods for quantitative diagnostics of such structures are urgently needed. Raman spectroscopy is an extremely useful tool for non-destructive studies of the phonon spectrum of SLs with a high spatial resolution. A significant number of publications are devoted to theoretical and experimental studies of the Raman spectrum of binary SLs caused by diffusion of interfaces [10,11,12,13,14,15,16]. However, most of them focus on GaAs/AlAs-based SLs. To our knowledge, there is only one theoretical study in which the effect of interface diffusion on the Raman scattering of light in short-period zinc-blende GaN/AlN SLs has been considered [17]. At the same time, GaN/AlN SLs with a wurtzite structure are the basis of all modern nitride-based optoelectronics. The type of crystal structure alters the Raman spectrum profoundly. The Raman spectrum of zinc-blende GaN/AlN SLs contains only two optical phonons, while the spectrum of wurtzite GaN/AlN SL contains six optical phonon lines, which is a favorable factor for the development of quantitative diagnostic methods. There are a number of works where Raman spectroscopy was used to quantitatively evaluate important structural parameters of short-period GaN/AlN SLs like their period, thickness of the GaN and AlN layers, sign and magnitude of strain in individual layers comprising SLs [18,19,20,21,22,23,24,25]. However, as far as we know, there are no studies yet on the possibility of estimating the degree of interface diffusion in wurtzite short-period GaN/AlN SLs from the data of Raman spectroscopy.

This paper presents the results of joint theoretical and experimental studies to extract from the Raman spectra of short-period wurtzite GaN/AlN SLs the sharpness of the interfaces between the GaN and AlN layers to further aid the optimization of such structures.

## 2. Experimental and Calculation Details

Short-period equal-thickness SLs (GaN)*_m_*/(AlN)*_m_*, where *m* is the number of monolayers; *m* = 4, 6, 8 with thickness of 1 ML = 0.2593 (0.2491) nm for GaN(AlN), respectively, were studied both experimentally and theoretically. Two techniques were employed for the SL growth, namely plasma-assisted molecular beam epitaxy (PA MBE) using a Compact 21T setup (Riber, Bezons, France) equipped with a N_2_-plasma source HD-25 (Oxford Appl. Res. Ltd., Oxfordshire, UK) and horizontal inductively heated single-wafer metal-organic vapor phase epitaxy (MOVPE) reactor Epiquip VP-50RP (Epigress, Lund, Sweden) re-designed by Submicron Heterostructures for the Microelectronics Research & Engineering Center of the Russian Academy of Science (St-Petersburg, Russia).

The Raman measurements were performed at room temperature using a T64000 (Horiba Jobin-Yvon, Lille, France) spectrometer equipped with a confocal microscope. The line at 532 nm (2.33 eV) of Nd: YAG laser (Torus, Laser Quantum, Inc., Edinburgh, UK) was used as the excitation source. All spectra were measured in backscattering geometry $z(xx)\bar{z}$; here, *z* is the direction of the 3-fold optical axis, and *x* and *y* are mutually orthogonal axes, which are arbitrarily oriented in the substrate plane. All experimental Raman spectra are shown upon subtracting the AlN buffer spectrum.

The *ab initio* calculations using the plane-wave pseudopotential method were carried out within the framework of density functional theory in the local density approximation (DFT-LDA) as realized in the ABINIT software package [26,27,28]. The phonon wave vectors and frequencies were obtained in the Г-point of the BZ within the density functional perturbation theory (DFPT) [29,30]. The Raman spectra of SLs with sharp interfaces were simulated from the Raman tensor, the components of which are the third-order total-energy derivatives (with respect to the electric field and the atomic displacements) calculated within the perturbation theory by applying the (2*n* + 1) theorem [31].

The SLs’ structures with diffuse interfaces were analyzed assuming that the cation sites in the interface regions are occupied randomly by both cations (Ga and Al), while keeping the given ratios in each atomic plane perpendicular to the *z*-axis. The Raman spectra simulations of such structures were performed using the random-element isodisplacement (REI) semi-empirical model [32], and using the SUPERCELL model [33] within the *ab initio* approach.

## 3. Results and Discussion

### 3.1. Growth

All the samples were grown on *c*-sapphire substrates. When using PA MBE, the growth of AlN buffer layers starts on annealed and nitrided substrates. As a next step, using migration-enhanced epitaxy at a substrate temperature of 780 °C, a nucleation layer with the thickness of 65 nm was formed on these substrates, and then, using metal-modulated epitaxy at the same temperature, the buffer layers with a thickness of about 300 nm were grown as described in detail in [34]. The SLs with the total thickness of about 500 nm were grown at relatively low growth temperatures of 690–700 °C at metal-enriched conditions with the F_III_/F_N2*_ flux ratio above 1, followed by periodic annealing of the excess metallic (Ga) phase in accordance with the approach described in [35]. Low growth temperatures were chosen to kinetically suppress Ga segregation, which usually manifests itself at the upper interface of GaN/AlN quantum wells of any thickness if they are grown at higher temperatures, which is associated with a significant difference in the binding energies of Ga and Al atoms with nitrogen [36]. Metal-enriched PA MBE growth conditions increase the surface mobility of both Group III atoms and nitrogen atoms due to the formation of a double adlayer of Group III atoms on the surface, which leads to the implementation of two-dimensional growth mechanisms of III-N layers even at relatively low substrate temperatures. In addition, the sharpness of the interfaces was provided by a high switching rate of Ga and Al fluxes (<0.3 s) incident on the substrate at the used ultralow working pressures (<2 × 10^−5^ Torr) of nitrogen in the growth chamber.

MOVPE structures were grown on (0001) sapphire substrates using trimethylgallium (TMGa), trimethylaluminum (TMAl), and ammonia (NH_3_) as precursors at 100 mbar reactor pressure. The thickness of the layers was controlled by the growth time. Only for the structure with the thinnest layers, the growth rate of AlN and GaN layers of the SL was also reduced. Growth was initiated with 20 min of growth of the 130 nm thick AlN buffer layer at 980 °C. Then, the reactor temperature was raised to 1030 °C, which is a standard temperature for GaN growth in our setup. In the SLs, the GaN layers were grown for 3–6 s with 3.5 SLM of NH_3_ flow using 5.5 SLM of nitrogen and 1 SLM of hydrogen as a carrier gas. The AlN layers in the SLs were grown for 10–24 s with 20 sccm of NH_3_ flow and 8.2 SLM of nitrogen as a carrier gas. Two ammonia supply lines and two carrier gas supply lines were used to realize this growth sequence. Due to a very small volume of our reactor, the gas composition switching occurs in much less than 1 s, and the above mentioned durations are long enough for the SL growth. It is known that low ammonia flow during AlN growth improves its quality, while GaN growth requires much higher ammonia flow. Switching of carrier gas composition as described above, was aimed at improving the sharpness of interfaces during the growth of AlN layers with low ammonia flow above GaN layers. In this case, the growth of structures conventionally using hydrogen as a carrier gas will inevitably lead to the destruction of the GaN surface and the formation of intermedia AlGaN layers due to the interaction of GaN with hydrogen, as described in [37].

### 3.2. Raman Spectra of PA MBE- and MOVPE-Grown SLs and REI Modeling

As an example, Figure 1 shows the Raman spectra of two short-period (GaN)_6_/(AlN)_6_ SLs, one of which was grown with PA MBE and the other with MOVPE. In the used scattering geometry $z\left(xx\right)\bar{z}$ the *E*(TO) and *A*_1_(LO) symmetry modes should be observed in the SLs’ spectra. In the spectral range from 550 to 650 cm^−1^, two optical modes of *E*(TO) symmetry appear, genetically related to the *E*_2_(high)- and *E*_1_-branches of the GaN and AlN bulk crystals [19]. It has been established that the *E*(TO) modes are localized in the constituent SL layers and can be used to obtain information about the individual characteristics of each layer forming the SL, for example, about the strain in GaN and AlN [36]. It can be seen that the FWHM of *E*(TO) phonon lines in the spectrum of the MOVPE-grown SL is much larger than the FWHM for the similar lines in the spectrum of the PA MBE-grown SL (for example, the FWHM for the highest intensity band, which corresponds to the *E*(TO) mode in the case of SLs grown by MOVPE is equal to Δω = 15.6 cm^−1^, which is much higher than the one for the SLs grown by PA MBE Δω = 4.6 cm^−1^). The significantly larger line widths in the Raman spectrum of the MOVPE-grown SLs unambiguously indicate larger structural disorder in their constituent layers compared to the PA MBE-grown SL layers.

The formation of GaN/AlN SLs induces the folding of the Brillouin zone (BZ) in the growth direction. As a result, the Raman spectra of SLs should contain a set of folded *A*_1_(LO) lines, which are associated with phonons belonging to two high-frequency branches (*A*_1_(LO) and *B*_1_(high)) of phonon dispersion in the Г—A direction of the BZ (see the inset in Figure 1). The ranges of dispersion of these modes extend from 735 to 890 cm^−1^ for bulk AlN and from 692 to 733 cm^−1^ for bulk GaN crystals [38,39]. Therefore, the four lines that are observed in the spectrum of each SL in the range above 692 cm^–1^ originate from folded *A*_1_(LO) phonons.

For PA MBE- and MOVPE-grown SLs, the spectra in Figure 1 differ most in the spectral range of *A*_1_(LO) phonons. The theory based on the assumption of an ideally sharp interface, that is, on a model in which there is no interlayer diffusion of cations, predicts for such phonons the character of standing waves strictly localized in GaN and AlN layers of SLs. As a result, the Raman spectra of SLs in the frequency range of such phonons contain two sets of lines in the frequency ranges from B_1_ to *A*_1_(LO) phonon modes of bulk materials.

The number and frequencies of the confined *A*_1_(LO) phonons are extremely sensitive to the variation of thickness of AlN layers. This is consistent with the large dispersion range of *A*_1_(LO) and *B*_1_(high) phonons of bulk AlN. By the number of such lines, one can estimate the thickness of the corresponding layers [35]. On the contrary, the dispersion of *A*_1_(LO) and *B*_1_(high) phonons in bulk GaN is nearly flat along the Г–A direction. As a result, the confined *A*_1_(LO) phonons in GaN layers occupy a narrow range of the spectrum, thus having a low sensitivity to changes in thickness of GaN layers.

Figure 2 shows the results of *ab initio* simulation of Raman spectrum of the (GaN)_6_/(AlN)_6_ SL in the spectral region of *A*_1_(LO) modes. The calculations were performed under the assumption of sharp interfaces between the SL layers; a detailed description of such calculations is given in [35]. The same figure shows the experimental Raman spectra for two (GaN)_6_/(AlN)_6_ SLs grown by PA MBE and MOVPE. As can be seen, a good agreement is observed between the *ab initio* simulated spectrum and the experimental spectrum obtained from the SL grown by PA MBE, which indicates the presence of sharp interfaces in the latter. However, the experimental spectrum for the SL grown by MOVPE differs significantly from the spectrum calculated *ab initio*. This finding, together with the significant broadening of the *E*(TO) lines noted above, prompted us to assume that there is a substantial interlayer diffusion of cations in the MOVPE-grown SLs, and thus their interfaces differ from the sharp ones.

In order to verify this hypothesis, we simulated the Raman spectra of SLs with diffuse interfaces. The diffusion of sharp interfaces in binary SLs implies the presence of regions, in which cation sites in each atomic plane perpendicular to the growth axis are randomly occupied by Ga and Al atoms, whereas their relative concentration Ga/Al is assumed to change linearly within the interface layer from 1/0 to 0/1 (and *vice versa*). Modeling the dynamics of such systems using *ab initio* calculations is a difficult task associated with the use of large supercells. Fortunately, within the framework of classical lattice dynamics based on the use of model interatomic potentials, the approach has been developed to simulate phonon states and vibrational spectra of crystals with random occupation of positions by different atoms (i.e., solid solutions) without special computational costs. The idea behind the method, called random-element isodisplacement (REI), dates back to [32]. Traditionally, the REI model was used for studying concentration dependence of the phonon frequencies in bulk solid solutions. For this purpose, it was successfully applied to the AlGaN alloys in a series of studies [40,41,42,43]. In the present work, we applied a similar approach to model the phonon states in binary GaN/AlN SLs with diffuse interfaces (for details see Appendix A). This approach is supported by the symmetry analysis of the interface regions.

To classify the phonon modes obtained by the REI method according to irreducible representations of space groups, we have proposed a symmetry description of (GaN)*_m_*/(AlN)*_n_* SLs with diffuse interface regions. For doing this, we start from the perfect SLs and then begin to replace atoms in the neighboring atomic planes by virtual Ga/Al atoms.

In perfect (GaN)*_m_*/(AlN)*_n_* (*m* = 2r, *n* = 2s) SLs with sharp interfaces, the symmetry of which is described by the space group *C*_3v_^1^ (*P*3*m*1), the Ga, Al, and N atoms are distributed over the symmetry positions 1b ($\frac{1}{3}$$\frac{2}{3}$z) and 1c ($\frac{2}{3}$$\frac{1}{3}$z) (Wyckoff positions). The coordinates of both positions contain a free parameter z; therefore, Ga (Al, N) atoms form a set of orbits (that is, systems of points that transform into each other under the action of space group symmetry operations). In the space group *C*_3v_^1^ (*P*3*m*1), each orbit from the set corresponds to atoms located in one plane perpendicular to the z axis and characterized by the z_i_ coordinate. In the case under consideration, the multiplicity of 1b and 1c positions is equal to 1, so each orbit contains only one point per primitive unit cell.

In the family of (GaN)*_m_*/(AlN)*_n_* SLs with *m* = 2r, *n* = 2s, half of the Ga ($\frac{m}{2}$) atoms in a primitive unit cell are located in planes perpendicular to the *z* axis and are characterized by coordinates 
($\frac{1}{3}$$\frac{2}{3}$z*_i_*) (*i* = 1, 2, …$\frac{m}{2}$.), where z*_i_* = $\frac{2\left(i-1\right)}{m+n}$ (in units of the SL translation vector along the *z* axis), and the second half is characterized by the coordinates ($\frac{2}{3}$$\frac{1}{3}$z*_i_*+$\frac{1}{m+n}$). Thus, the distance between the nearest neighboring Ga atomic planes in a primitive cell, belonging to different orbits, turns out to be equal to $\frac{1}{m+n}$. In turn, one half of the Al ($\frac{n}{2}$) atoms is characterized by coordinates ($\frac{1}{3}$$\frac{2}{3}$z*_j_* + $\frac{m}{m+n}$) (*j* = 1, 2, …$\frac{n}{2}$), where z*_j_* = $\frac{2\left(j-1\right)}{m+n}$, and the second half by coordinates ($\frac{2}{3}$$\frac{1}{3}$z*_i_* + $\frac{m}{m+n}$ + $\frac{1}{m+n}$).

Therefore, we can propose a symmetry model of the diffuse interfaces in SLs, which is consistent with the REI method, which implies the replacement of all cations near the interface located in the same plane near the interface, and thus belonging to the same orbit, by virtual atoms *c*Ga/(1 − *c*)Al with an increasing fraction *c* (or 1 − *c*) of Ga (or Al) atom while moving away from the interfaces into the center of corresponding GaN (or AlN) layers which constitute the SL. In this case, the replacement of all atoms of one orbit by any identical atoms does not change either the space symmetry group of the lattice or its structural type. This will allow us to use the selection rules derived for perfect (GaN)*_m_*/(AlN)*_n_* SLs with sharp interfaces also for (GaN)*_m_*/(AlN)*_n_* SLs having diffuse interfaces.

Using the approach described above, we have calculated the phonon spectra of several SLs with different periods and different degrees of interface diffusion. Note that, within the framework of the REI model used, at any interface diffusion profile, the SL symmetry is conserved as shown above. As a result, the BZ-center phonons in such structures can be divided into *A*-phonons, in which atoms oscillate along the *C*_3_ symmetry axis, and *E*-phonons, in which atoms oscillate in the interface plane. When modeling the Raman spectra, special attention was paid to the $z\left(xx\right)\bar{z}$ scattering geometry. In this geometry, the phonons of *E*(TO) and *A*_1_(LO) symmetry are the Raman-active ones. In this paper, we studied the effect of the interface diffusion on the *A*_1_(LO) phonon spectrum in GaN/AlN SLs with equal layer thicknesses and with a total period N varying from 8 to 16 monolayers. The thickness of both interfaces was assumed to be the same. To characterize the interface diffusion numerically, we used the parameter *I*, defined as the number of successive cation planes with partial occupancies of Wyckoff positions by Ga and Al atoms.

As an example, Figure 3 shows the results obtained using the REI model for the (GaN)_6_/(AlN)_6_ SL. The upper graph in Figure 3b corresponds to (GaN)_6_/(AlN)_6_ SL with a sharp interface (*I* = 0). Along with the calculated Raman spectra, Figure 3b shows the calculated mode frequency distribution. In the $z\left(xx\right)\bar{z}$ spectrum of a (GaN)*_m_*/(AlN)*_n_* SL, we can expect *m* lines in the range within the *B*_1_–*A*_1_(LO)_GaN_ and *n* lines in the range *B*_1_–*A*_1_(LO)_AlN_. It is seen that for the (GaN)_6_/(AlN)_6_ SL, the calculation predicts 6 phonon modes in the range of 692–733 cm^−1^ and 6 phonon modes in the range of 735–890 cm^−1^. As noted above, in (GaN)*_m_*/(AlN)*_n_* SLs with sharp interfaces, the *A*_1_(LO) modes are standing waves confined in separate layers of the SL. It was shown in [35] that only half of them will give noticeable spectral peaks (this is determined by the parity of the harmonic). The most intense peak will correspond to a harmonic with a wavelength equal to half the thickness of the corresponding layer. Its frequency will be close to *A*_1_(LO) of a bulk crystal. Indeed, in the calculated Raman spectrum, two lines dominate at frequencies close to *A*_1_(LO)_GaN_ and *A*_1_(LO)_AlN_. Due to the narrowness of the *B*_1_–*A*_1_(LO)_GaN_ range, it is difficult to estimate the number of intense modes in this part of the Raman spectrum; however, it is clearly seen that in the range *B*_1_–*A*_1_(LO)_AlN_ for (GaN)_6_/(AlN)_6_ SL, the calculation predicts exactly three intense lines. Note that the spectra calculated within the REI model under the assumption of a sharp interface (*I* = 0) are in a very good agreement with the spectra calculated *ab initio* under the same assumption of a sharp interface. This gives reason to believe that the REI model should adequately reflect the features of SLs with diffuse interfaces.

Figure 3b with *I* = 2–6 shows what changes in the spectrum can be expected if the diffusion of interfaces increases (i.e., the parameter *I* increases). With increasing *I,* the total number of spectral lines does not change, but the frequency interval occupied by the *A*_1_(LO) line group narrows. Namely, the frequency of the *A*_1_(LO)_GaN_ line increases, and the frequency of the line *A*_1_(LO)_AlN_ decreases. Moreover, the intensity of the former decreases significantly, while the intensity of the latter decreases very little.

In an effort to understand the nature of the spectra evolution observed upon the interface broadening, we performed an analysis of the atomic displacement patterns for SL (GaN)_6_/(AlN)_6_ with varying degrees of interface diffusion. Let us consider atomic displacements in two modes close to *A*_1_(LO)_GaN_ and *A*_1_(LO)_AlN_. The calculated profiles of these vibrations are shown in Figure 4.

In general, the analysis of the vibration modes shown in Figure 4 confirms our assumption about the nature of the localization of vibrations: the dominant line of the first series (Figure 4a) corresponds to the mode with the largest displacements in the GaN layers, and the dominant line of the second series (Figure 4b) corresponds to the mode with the largest displacements in the AlN layers. Such localization is the strongest for the SLs with sharp interfaces (modes *I* = 0 (735 cm^−1^) and *I* = 0 (886 cm^−1^)). However, while the *I* = 0 (886 cm^−1^) mode is almost completely localized in the AlN layers, the localization of the *I* = 0 (735 cm^−1^) mode in GaN layers is not complete: displacements of N atoms are noticeable in AlN layers. Moreover, these displacements are in opposite phases, which is characteristic of the *B*_1_ mode of a bulk crystal. Thus, the dominant line of the first series is not a pure *A*_1_(LO) mode localized in the GaN layers, but a mixed *A*_1_(LO)_GaN_ + *B*_1AlN_ vibration. With increasing interdiffusion of the interface, this mixing also increases, which is the reason for a decrease in the intensity of this line.

Let us discuss this mixed vibration in detail. It is seen in Figure 4a (panel *I* = 0 (735 cm^−1^)) that among cations, only the Ga atoms are displaced in the GaN layers (all red bars are downward) in this mode. In Figure 4a, panels *I* = 2 (741 cm^−1^), *I* = 4 (751 cm^−1^), and *I* = 6 (760 cm^−1^) show that, as the diffusion of the interface increases, Al atoms begin to participate more and more in the displacements (green bars become comparable with red ones). As a result, the mass of the cation averaged over the layer decreases, the reduced mass decreases, and the vibration frequency increases. This can explain the shift of the GaN-like line to the high frequencies.

Let us discuss next the dominant line of the second series, i.e., the *A*_1_(LO)_AlN_ line. Figure 4b shows that this mode remains localized in AlN layers at any interface diffusion (taking into account its frequency, it has no counterpart to be mixed with in GaN layers). This can explain the weak change in the intensity of this mode. While talking about the frequency of this line, two circumstances should be noted. First, as the interface diffusion increases, the range of localization of this mode narrows. While in the *I* = 0 (886 cm^−1^) mode there are noticeable amplitudes of six N atoms, only five N atoms contribute noticeably to the *I =* 6 (867 cm^−1^) mode. Narrowing of the localization interval leads to a decrease in the wavelength of the standing harmonic or to an increase in the wave vector of the confined phonon and, consequently, to a decrease in its frequency (see the inset in Figure 1). Second, Figure 3a shows that an increase of the interface diffusion is accompanied by an increase in the concentration of Ga cations in the AlN layers. We recall that the effective charge (1.14*e*) of Ga cations is noticeably lower than that (1.27*e*) of Al cations. Thus, an increase in the interface diffusion leads to a decrease in the average effective charge in the AlN layers. A decrease in the effective charge leads to a decrease in LO-TO splitting, that is, to a decrease in the LO-mode frequency. These two factors help to understand the low-frequency shift of the *A*_1_(LO)_AlN_ line.

A similar argument can be put forward for the explanation of the *A*_1_(LO)_GaN_ line shift to the high *A*_1_(LO)_GaN_ frequencies, because penetration of Al cations into GaN layers increases the effective charge.

Figure 5 shows the results of *ab initio* simulation of Raman spectra of (GaN)_8_/(AlN)_8_ and (GaN)_4_/(AlN)_4_ SLs in the spectral range of *A*_1_(LO) modes. The calculations were performed under the assumption of sharp interfaces between the SLs layers. The same figure shows the experimental Raman spectra for two sets of (GaN)_8_/(AlN)_8_ and (GaN)_4_/(AlN)_4_ SLs grown by PA MBE or MOVPE. As can be seen, a good agreement is observed between the *ab initio* simulated spectra and the experimental ones obtained for the SLs grown by PA MBE, which indicates the presence of sharp interfaces in the latter. However, the experimental spectra for the SLs grown by MOVPE differ significantly from the spectra calculated *ab initio*.

Using the RIE method described in detail above, we calculated the phonon spectra of (GaN)_8_/(AlN)_8_ and (GaN)_4_/(AlN)_4_ SLs with different degrees of interface diffusion. The model profiles of the interfaces and the results of calculations of the Raman spectra in the range of the *A*_1_(LO) phonons are shown in Figure 6.

Analysis of the results presented in Figure 4, Figure 5 and Figure 6 allows us to make the following conclusion. As the degree of interface diffusion increases, the *A*_1_(LO) modes localized in the GaN and AlN layers begin to mix. This leads to significant changes in the Raman spectrum. The intensity of the *A*_1_(LO)_GaN_ line decreases significantly, while the intensity of the *A*_1_(LO)_AlN_ line decreases very little. In addition, the intensities of other *A*_1_(LO) harmonics increase. Along with this effect, the frequency interval occupied by the *A*_1_(LO) lines gradually narrows, while the frequency of the *A*_1_(LO)_GaN_ line increases and the frequency of the *A*_1_(LO)_AlN_ line decreases.

Note that the similar behavior of the *A*_1_(LO) lines was reported in [17], where the results of calculations were presented for the Raman spectra of a zinc-blende (GaN)_8_/(AlN)_8_ SLs and the interface diffusion, which was simulated by the Al_x_Ga_1−x_N solid solution with a thickness of one to three monolayers.

Qualitatively, the results of calculations within the framework of the REI model reflect all the main trends in the spectra of (GaN)*_m_*/(AlN)*_m_*SLs grown by the MOVPE. To verify this, let us compare the experimental spectra with the results of calculations for *I* = 4, 6, and 6 for *m* = 4, 6, and 8, respectively, as is done in Figure 7. It can be seen that both in the calculation and in the experiment, the same changes are observed: as the SL period decreases, the number of lines decreases, the highest-frequency line shifts to the lower frequencies, and the lowest-frequency line shifts to the higher frequencies. In addition, the calculation correctly reproduces a significant decrease in the intensity of the line at *A*_1_(LO)_GaN_~750 cm^−1^ and the equalization of the intensities of all *A*_1_(LO) modes. These results confirm the validity of the assumption about the interface diffusion in MOVPE-grown SLs and make it possible to estimate the interface thickness in each of the studied SLs by choosing the interface diffusion parameter *I* that best reproduces the observed spectrum.

Additional calculations show that, with a significant interface diffusion, the SL Raman spectrum approaches the spectrum of a digital solid solution: all *A*_1_(LO) lines merge into one peak corresponding to the *A*_1_(LO) mode of the solid solution [39].

In conclusion of this section, we present Figure 8, which shows the spectra of SLs grown by MOVPE with a period varying over a very wide range of values. All samples were grown using AlN buffer. Growth details were similar to those described in [23]. It can be seen that with an increase in the SL period, the intensity of the *A*_1_(LO)_GaN_ (~730 cm^−1^) mode increases, and the Raman spectrum becomes similar to the *ab initio* calculated spectrum of an SL with sharp interfaces. This is a quite predictable result, indicating an increase in the degree of localization of the *A*_1_(LO)_GaN_ mode with an increase in the thickness GaN layer in the SL, caused by a decrease in the influence of the interface diffusion, which causes it to mix with the *B*_1AlN_ mode. The observed trend in the spectra presented in Figure 8 indicates that the equality of the intensities of all *A*_1_(LO) lines is a direct indication of the presence of a diffuse interface, while the degree of interface diffusion can be estimated from the width of the spectral range occupied by the *A*_1_(LO) phonon lines.

### 3.3. Ab Initio Modeling a Diffuse Interface Using the SUPERCELL Method

In order to support the results of semi-empirical calculations, a more advanced and completely *ab initio* approach was undertaken to describe the SLs with diffuse interfaces. The appropriate method to describe a structure derived from the parent (initial) high-symmetry one with the cation sites being occupied by two chemical elements with a given ratio is the well-known SUPERCELL one [33,44,45,46]. This method suggests the construction of a supercell having lower symmetry with full occupancy of the supercell sites while conserving the ratio of both cations.

For example, in the case of (GaN)_4_/(AlN)_4_ SLs, the size of the supercell can be chosen as 2 × 2 × 1 (the primitive unit cell increased four times). In such a supercell, the interface layer is constructed by two disordered layers with atomic positions in the unit cell randomly occupied by atoms Ga and Al with a given ratio. The construction of the supercell replaces random occupancy of the cation sites by different atomic configurations. The example of such a supercell is shown in Figure 9.

For the 2 × 2 × 1 supercell, the condition of equal concentrations of two cation atoms leads to restriction on atomic configurations of cations occupying the corresponding sites in the interface layers. The allowed values are given in Table 1.

There are 36 different variants of atomic configurations for the interface type I, with space symmetry of such structures lowering down to monoclinic, namely *Pm* (12 structures), and triclinic ones, *P*1 (24 structures).

In the case of interface type II, for supercells with non-equal concentrations of Ga and Al cations, there are only 16 variants of atomic configurations of the interface layers. Moreover, for 4 supercells, the symmetry is conserved to be trigonal (space group *P*321), whereas for 12 supercells it lowers to the monoclinic one (space group *Pm*).

There are 64 atoms in the 2 × 2 × 1 supercell, which is twice as much as the unit cell of the (GaN)_8_/(AlN)_8_ SL. Accordingly, *ab initio* calculations of non-linear properties of such structures are very resource demanding. Taking into account the limitations on the supercell size, the *ab initio* calculations were performed for supercells only twice as much as the initial unit cell. The unit cells are related by the following equation:(a’,b’,c’)=(a,b,c)P,
where (a,b,c) is the basis of the initial unit cell, (a’, b’, c’) is the basis of the supercell, and *P* is the (3 × 3) transformation matrix.

To find a proper transformation matrix, the program CELLSUB [47], which is a part of BCS [48] was used. As a result, the subgroups with a given index k = 6 were obtained (Table 2).

As far as the lattice parameter *c* in GaN/AlN SLs should be conserved, variant 2 in Table 2 is not relevant. Variants 1 and 3 are equivalent and, accordingly, variant 1 was used in construction of supercells.

According to the results of the WYCKSPLIT program (Table 3), the WPs 1*c* and 1*b* of the parent group *P*3*m*1 are split into WPs 1*a* and 1*b* of the supercell with space group *Pm*.

The calculations were performed for supercells constructed for the (GaN)_4_/(AlN)_4_ and (GaN)_6_/(AlN)_6_ SLs using the transformation matrix discussed above. There are two types of different distorted interface layer supercells, with 32 and 48 atoms for (GaN)_4_/(AlN)_4_ and (GaN)_6_/(AlN)_6_ SLs, respectively. Atomic positions of the supercells were optimized with lattice parameters being fixed. The total Raman spectra of SLs with distorted interface layer were obtained as a sum of Raman spectra of both types of supercells.

The *ab initio* calculated Raman spectra shown in Figure 10a were performed under the assumption of sharp interfaces in (GaN)*_m_*/(AlN)*_m_*(*m* = 4, 6). All main features of the calculated spectra are in good agreement with the experimental spectra obtained on SLs with the same parameters, grown by PA MBE (Figure 2b and Figure 5b). Namely, the band at ~730 cm^−1^, which corresponds to the *A*_1_(LO) phonon mode localized in GaN layers, dominates with approximately the same intensity for both *m* = 4 and *m* = 6. Besides, there are two lines for *m* = 4 and three lines for *m* = 6 in the spectral range *B*_1_–*A*_1_(LO)_AlN_, which agrees with the results presented in [35] for (GaN)*_m_*/(AlN)*_m_*SLs with sharp interfaces.

In turn, the Raman spectra shown in Figure 10b were obtained for (GaN)_4_/(AlN)_4_ and (GaN)_6_/(AlN)_6_ SLs with diffuse interfaces using the SUPERCELL model within the *ab initio* approach. In general, the results presented in this Figure reproduce all main features of the experimental Raman spectra of SLs grown by MOVPE (Figure 2c and Figure 5c). The spectra do not contain a dominant line corresponding to the *A*_1_(LO)_GaN_ mode, but reveal wide and complex spectral feature with number of peaks increasing along with layer thickness. This behavior is in good agreement with the results obtained by the REI method and indicates a significant interlayer diffusion of cations in SLs grown by the MOVPE.

## 4. Summary and Conclusions

Summing up, we have performed complex theoretical and experimental studies aimed at identifying features in the Raman spectra, which can be used for evaluation of the interface quality between GaN and AlN layers in wurtzite short-period GaN/AlN SLs. Experimental Raman spectra were obtained on the SLs grown by PA MBE and MOVPE. The spectra in the range of *A*_1_(LO) confined phonons were interpreted using theoretical Raman spectra obtained by *ab initio* calculations and in the framework of the REI model. A good agreement was observed between the simulated spectra for SLs with sharp interface and the experimental ones measured on the PA MBE-grown SLs. In order to demonstrate that there is a significant interlayer diffusion of cations in the MOVPE grown SLs, the Raman spectra of SLs with various degrees of interface diffusion have been simulated within the framework of the REI method. The diffuse interfaces were modeled by a Ga_x_Al_1−x_N solid solution, in which the compositional parameter *x* is a linear function of the z-coordinate along the growth axis of the planar heterostructure. It was found that the results of calculations within the framework of the REI model reflect with sufficient accuracy all the main trends in the experimental spectra of SLs grown by MOVPE. The more advanced SUPERCELL model takes into account all possible atomic configurations and their symmetries within each atomic monolayer in the interface region. This makes it possible to obtain the *ab initio* simulated Raman spectra of (GaN)*_m_*/(AlN)*_m_* SLs with diffuse interfaces. The *ab initio* approach supports our assumption that interfaces in the PA MBE-grown SLs are close to sharp ones, whereas those in the MOVPE-grown SLs are significantly diffuse. It is shown that by fitting the interface diffusion parameter that allows reproducing the observed spectrum, it is possible to estimate the interface diffusion in each of the studied SLs. This opens up new possibilities for the analysis of the structural characteristics of short-period GaN/AlN SLs using Raman spectroscopy. The results of the comprehensive studies can be used to optimize the parameters of the growth process in order to form structurally perfect short-period GaN/AlN SLs.

## Figures and Tables

**Figure 1 nanomaterials-11-02396-f001:**
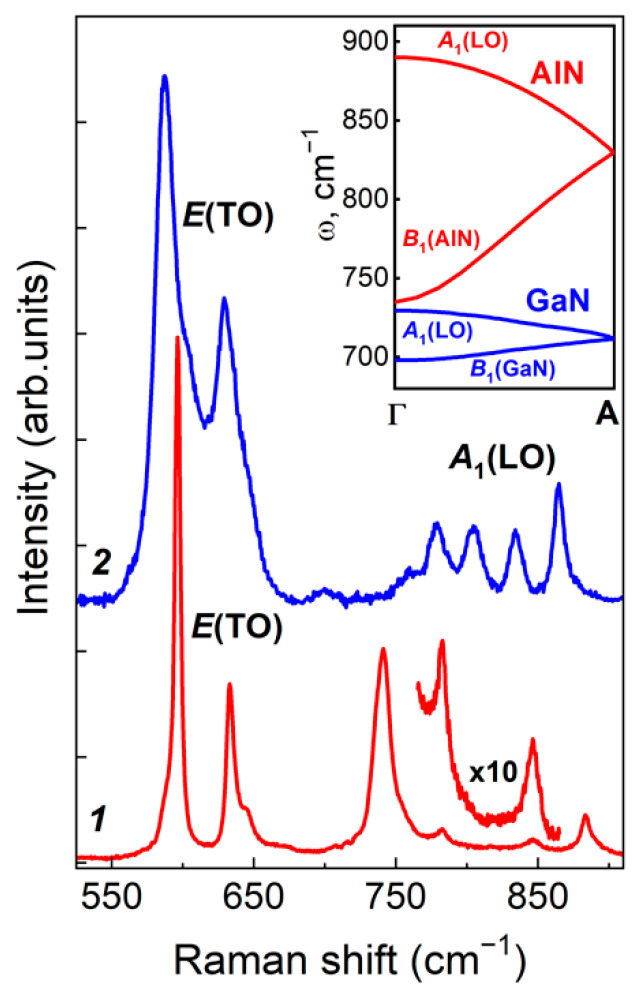
Experimental $z\left(xx\right)\bar{z}\;$ Raman spectra of the (GaN)_6_/(AlN)_6_ SLs in the range of *E*(TO) and *A*_1_(LO) modes. The SLs were grown by PA MBE (*1*) and MOVPE (*2*). The inset shows the dispersion range of *A*_1_(LO) and *B*_1_(high) phonons of bulk AlN and GaN along the Г—A direction of the BZ [38,39].

**Figure 2 nanomaterials-11-02396-f002:**
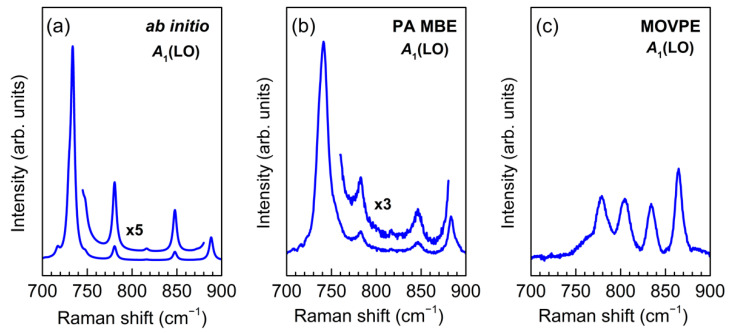
*Ab initio* simulated (**a**) and experimental (**b**,**c**) $z\left(xx\right)\bar{z}\;$ Raman spectra of (GaN)_6_/(AlN)_6_ SLs in the range of *A*_1_(LO) modes. The SLs were grown by PA MBE (**b**) and MOVPE (**c**).

**Figure 3 nanomaterials-11-02396-f003:**
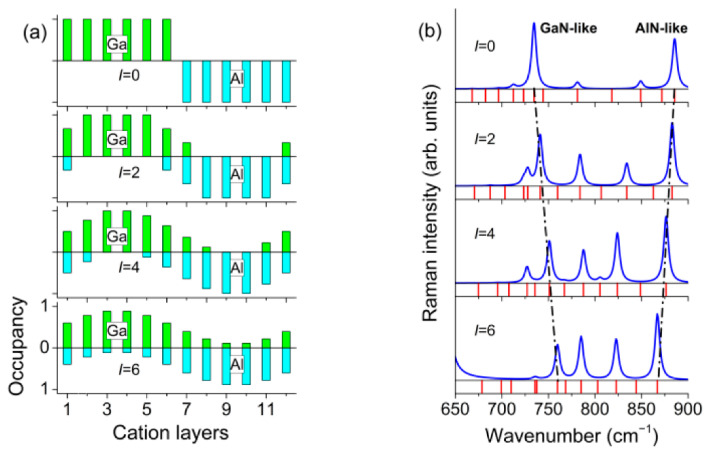
Cation concentration profiles of (GaN)_6_/(AlN)_6_ SLs with different degrees of interface diffusion (**a**). $z\left(xx\right)\bar{z}\;$ Raman spectra of (GaN)_6_/(AlN)_6_ SLs calculated within the REI model (blue lines) and the frequency distribution of *A*_1_(LO) modes (red lines) (**b**).

**Figure 4 nanomaterials-11-02396-f004:**
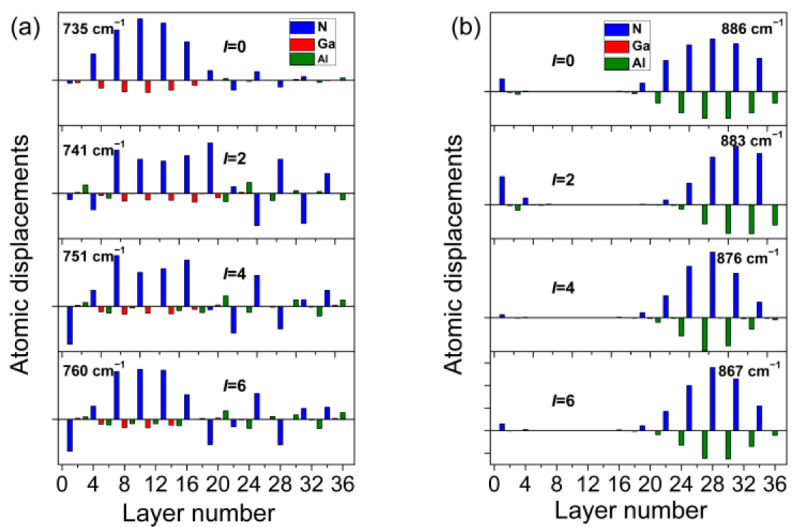
Atomic displacement patterns of *A*_1_(LO)_GaN_ (**a**) and *A*_1_(LO)_AlN_ (**b**) modes for (GaN)_6_/(AlN)_6_ SL.

**Figure 5 nanomaterials-11-02396-f005:**
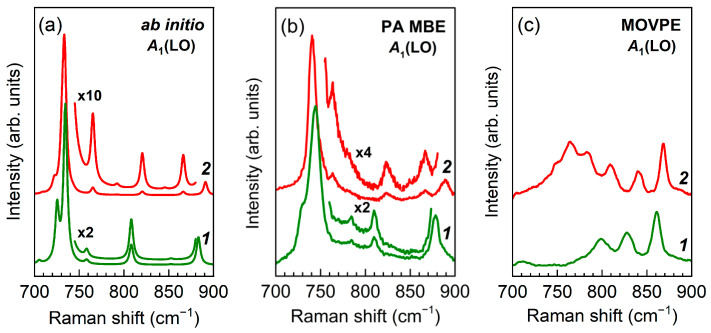
Calculated (**a**), and experimental (**b**,**c**) $z\left(xx\right)\bar{z}\;$ Raman spectra of (GaN)*_m_*/(AlN)*_m_* SLs in the range of *A*_1_(LO) modes: (*1*) *m* = 4, (*2*) *m* = 8, obtained upon subtracting the AlN buffer spectrum. The experimentally studied SLs were grown by PA MBE (**b**) and MOVPE (**c**).

**Figure 6 nanomaterials-11-02396-f006:**
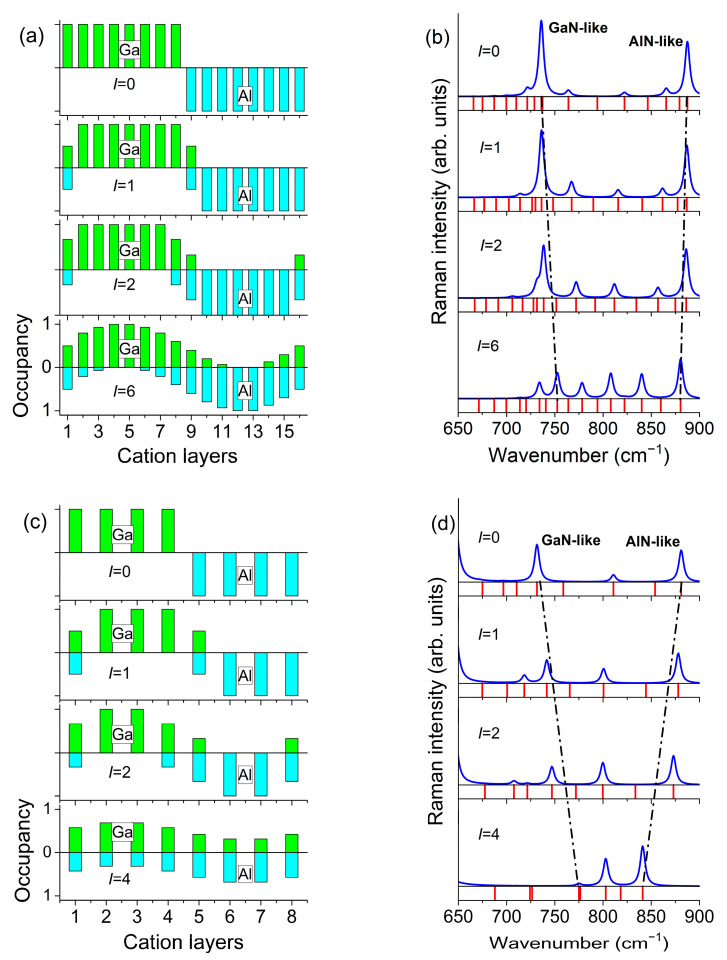
Cation concentration profiles of (GaN)*_m_*/(AlN)*_m_*SLs with different degree of interface diffusion: (**a**) *m* = 8, (**c**) *m* = 4. Calculated $z\left(xx\right)\bar{z}\;$ Raman spectra of (GaN)*_m_*/(AlN)*_m_*SLs (blue lines) and frequency distribution of *A*_1_(LO) modes (red lines): (**b**) *m* = 8, (**d**) *m* = 4.

**Figure 7 nanomaterials-11-02396-f007:**
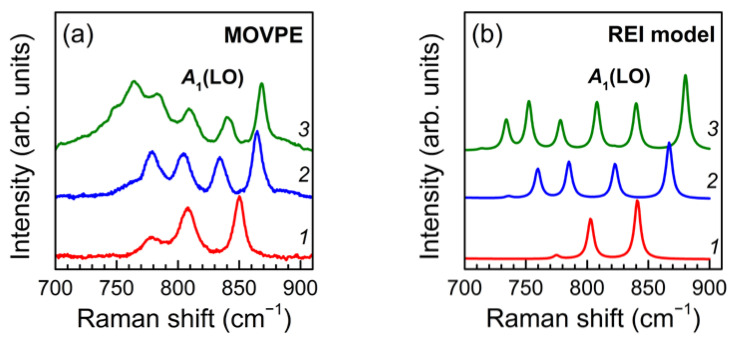
Experimental $z\left(xx\right)\bar{z}$ Raman spectra of (GaN)*_m_*/(AlN)*_m_*SLs grown by MOVPE: (*1*) *m* = 4; (*2*) *m* = 6; (*3*) *m* = 8 (**a**) in comparison with the spectra calculated for the same SLs within the REI model with *I* = 4, 6, and 6 for *m* = 4, 6, and 8, respectively (**b**).

**Figure 8 nanomaterials-11-02396-f008:**
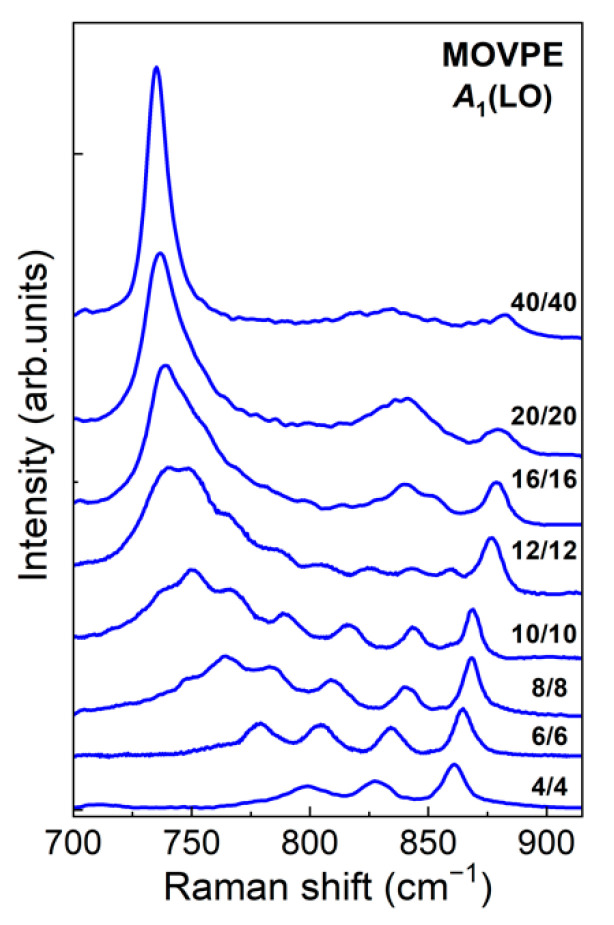
Experimental $z\left(xx\right)\bar{z}$ Raman spectra of (GaN)*_m_*/(AlN)*_m_* SLs with different periods.

**Figure 9 nanomaterials-11-02396-f009:**
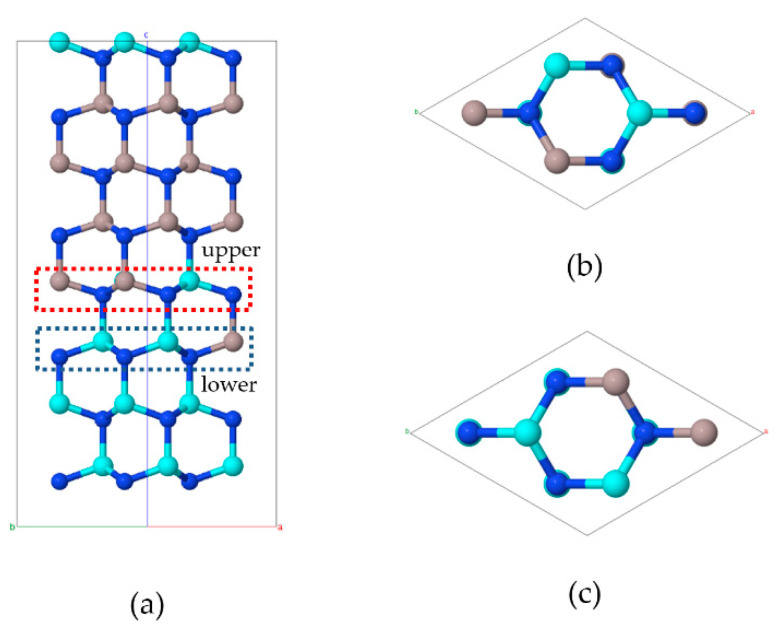
The [100] (**a**) and [001] (**b**,**c**) views of the supercell of a Ga(Al)N layer of the (GaN)_4_/(AlN)_4_ SL. Right panel: a view of the upper interface layer (**b**), a view of the lower interface layer (**c**). Blue, cyan, and brown balls are N, Ga, and Al atoms, respectively.

**Figure 10 nanomaterials-11-02396-f010:**
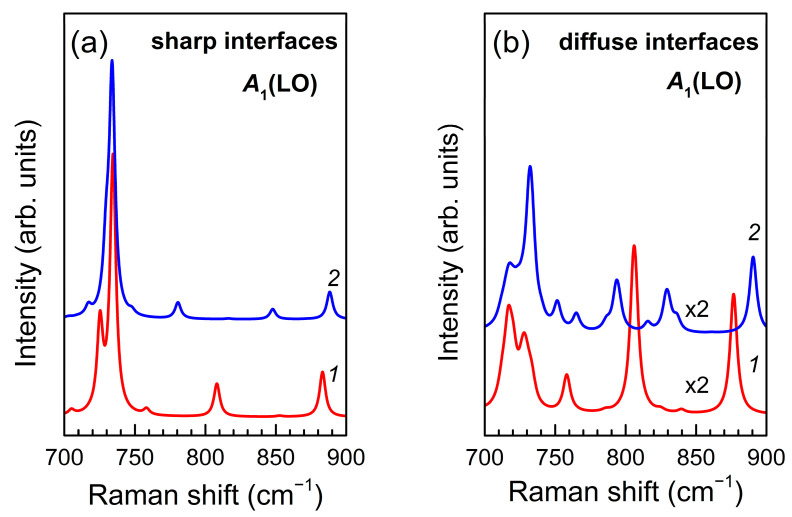
*Ab initio* simulated Raman spectra of (GaN)*_m_*/(AlN)*_m_* SLs with the sharp interfaces (**a**) and spectra of SLs with diffuse interface layers (**b**). Red line (*1*) corresponds to *m* = 4, blue line (*2*) corresponds to *m* = 6.

**Table 1 nanomaterials-11-02396-t001:** Two cases of partial occupancy values of cation atomic positions in interface layers of the parent high-symmetry structure.

	Interface Type I	Interface Type II
Upper layer Al (Ga)	0.5(0.5)	0.75(0.25)
Lower layer Al (Ga)	0.5(0.5)	0.25(0.75)

**Table 2 nanomaterials-11-02396-t002:** Group-subgroup relations and transformation matrices.

Variant	Chain [Indices]	Chain with HM Symbols	Transformation Matrix	**Identical**
1	156 008 006 [3 2]	P3m1>Cm>Pm	$\left(\begin{array}{cccc} 2 & 0 & 0 & 0\\ 1 & 1 & 0 & 0\\ 0 & 0 & 1 & 0 \end{array}\right)$	To group 1
2	156 008 006 [3 2]	P3m1>Cm>Pm	$\left(\begin{array}{cccc} 1 & -1 & -1 & 0\\ -1 & -1 & 1 & 0\\ -1 & 0 & 0 & 0 \end{array}\right)$	To group 2
3	156 008 006 [3 2]	P3m1>Cm>Pm	$\left(\begin{array}{cccc} -1 & 1 & 0 & 0\\ -2 & 0 & 0 & 0\\ 0 & 0 & 1 & 0 \end{array}\right)$	To group 3

**Table 3 nanomaterials-11-02396-t003:** Wyckoff position splitting schemes for the *P*3*m*1 >
*P*_m_ pair.

Variant	Wyckoff Positions	
	Group	Subgroup
1	1c	1a 1b
2	1b	1b 1a

## Data Availability

The data presented in this study are available on request from the corresponding author.

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
