# Peer review of "The Effect of Interface Diffusion on Raman Spectra of Wurtzite Short-Period GaN/AlN Superlattices"

_nanomaterials, 2021, doi:10.3390/nano11092396_

Round 1

Reviewer 1 Report

as attached file

Author Response

Dear Reviewer,

Reviewer 2 Report

In this manuscript, the authors performed complex theoretical and experimental studies aimed at identifying features in the Raman spectra, which can be used for evaluation of the interface quality between GaN and AlN layers in wurtzite short-period GaN/AlN superlattices. They simulated the Raman spectra for SLs with sharp interfaces and with different degree of interface diffusion by ab initio calculations and within the frame of random-element isodisplacement model. The results supports the assumption that interfaces in the PA MBE-grown SLs are close to sharp ones, whereas those in the MOVPE-grown SLs are significantly diffused. It is shown that by fitting the interface diffusion parameter that allows reproducing the observed spectrum, it is possible to estimate the interface diffusion in each of the studied SLs. The work is systematic and comprehensive. The manuscript is well organized and written. The proposed model and analysis method about Raman spectra can help other researchers to study the interface between GaN and AlN layers. I can recommend to accept it for publication.

Author Response

Dear Reviewer,
